# Medical Image Segmentation Using Automatic Optimized U-Net Architecture Based on Genetic Algorithm

**DOI:** 10.3390/jpm13091298

**Published:** 2023-08-25

**Authors:** Mohammed Khouy, Younes Jabrane, Mustapha Ameur, Amir Hajjam El Hassani

**Affiliations:** 1MSC Laboratory, Cadi Ayyad University, Marrakech 40000, Morocco; mohammed.khouy@ced.uca.ma (M.K.); y.jabrane@uca.ma (Y.J.); m.ameur@uca.ma (M.A.); 2Nanomedicine Imagery & Therapeutics Laboratory, EA4662—Bourgogne-Franche-Comté University, University of Technologie of Belfort Montbéliard, CEDEX, 90010 Belfort, France

**Keywords:** medical image segmentation, convolutional neural networks (CNNs), U-Net, genetic algorithms (GAs)

## Abstract

Image segmentation is a crucial aspect of clinical decision making in medicine, and as such, it has greatly enhanced the sustainability of medical care. Consequently, biomedical image segmentation has become a prominent research area in the field of computer vision. With the advent of deep learning, many manual design-based methods have been proposed and have shown promising results in achieving state-of-the-art performance in biomedical image segmentation. However, these methods often require significant expert knowledge and have an enormous number of parameters, necessitating substantial computational resources. Thus, this paper proposes a new approach called GA-UNet, which employs genetic algorithms to automatically design a U-shape convolution neural network with good performance while minimizing the complexity of its architecture-based parameters, thereby addressing the above challenges. The proposed GA-UNet is evaluated on three datasets: lung image segmentation, cell nuclei segmentation in microscope images (DSB 2018), and liver image segmentation. Interestingly, our experimental results demonstrate that the proposed method achieves competitive performance with a smaller architecture and fewer parameters than the original U-Net model. It achieves an accuracy of 98.78% for lung image segmentation, 95.96% for cell nuclei segmentation in microscope images (DSB 2018), and 98.58% for liver image segmentation by using merely 0.24%, 0.48%, and 0.67% of the number of parameters in the original U-Net architecture for the lung image segmentation dataset, the DSB 2018 dataset, and the liver image segmentation dataset, respectively. This reduction in complexity makes our proposed approach, GA-UNet, a more viable option for deployment in resource-limited environments or real-world implementations that demand more efficient and faster inference times.

## 1. Introduction

Medical image analysis is the initial step in medical image processing, which makes images more understandable and increases diagnostic effectiveness [1]. Medical image segmentation is an essential and critical step in the field of biomedical image processing, and it has significantly improved the sustainability of medical care [2]. It currently has a substantial study direction in the area of computer vision [1]. In some medical applications, it is desired to classify image pixels into distinct regions, such as bones and blood vessels. In other applications, searching for pathological regions, such as cancer or tissue deformities, is more applicable [3]. Additionally, to provide essential details regarding the size and volumes of body organs, a crucial task in medical image segmentation is identifying and removing redundant pixels or undesired background regions.

Several traditional machine learning and image processing approaches were used on the histogram characteristics [4], including region cut-based segmentation [5] and the segmentation method based on edge and region [6]. At present, image segmentation techniques are progressing more quickly and precisely. However, segmentation approaches that utilize deep learning (DL) have become very popular in recent years in the field of medical image segmentation, localization, and detection [2]. Deep learning techniques provide several benefits over traditional machine learning and computer vision techniques regarding segmentation accuracy and speed.

Deep learning approaches are known as universal learning approaches that allow a single model to be efficiently used in a variety of medical imaging modalities, such as magnetic resonance imaging (MRI), computed tomography (CT), and X-ray. According to [2,7], most publications have been published on segmentation tasks in various medical imaging modalities. With the rapid development of deep learning models, deep convolutional neural networks (DCNNs) [8] have attained great success in a broad range of computer vision, such as image classification and image segmentation, and they provide state-of-the-art performances in medical image segmentation [9]. CNNs have robust methodologies that allow image segmentation to be treated as semantic segmentation, which refers to the ability to understand an image at the pixel level by assigning a class label to each pixel of the image [10]. In [11], the researchers introduced a fully convolutional network (FCN), which is the foundation of most contemporary techniques for semantic segmentation. For biomedical image segmentation, the researchers suggested an encoder–decoder network of fully convolutional networks called U-Net [12]. Due to U-Net architecture’s excellent performance, since it was proposed in 2015, many scientists have adopted it as the backbone of their models. At present, U-Net is a popular network for biomedical image segmentation, and many variants of it exist, including Unet++ [13], AttentionU-Net [14] and MultiResUNet [15]. The U-Net model structure consists of two paths: contracting and expansive. The former is the contracting path, also known as the encoder or down-sampling; it is similar to a regular convolution network and gives the classification information. The latter is the expansive path, also known as the decoder or up-sampling, which contains up-convolutions and concatenations, which enable the network to learn localized classification information [16]. The concatenation in U-Net architecture, called skip connection [17], combines the information of the two paths in the U-Net architecture.

These models are very sophisticated and demand more GPU power, which makes them computationally expensive and requires substantial expertise. Furthermore, these neural architectures are manually designed and have a propensity for having many parameters, making them prone to overfitting when utilizing insufficient training data, and they generally have a computational complexity [18]. Therefore, it is not trivial to manually design the architecture of these models and select their appropriate hyperparameters based on past design experience.

In this study, we propose an automated design method for a state-of-the-art convolutional neural network, called GA-UNet, which is based on the genetic algorithm (GA). This is a metaheuristic method that generates a CNN architecture with a U-shape and automatically selects its hyperparameters that can achieve competitive performance with much shorter architecture compared with the original U-Net [12] architecture, which was manually designed. The proposed GA-UNet starts by designing a search space and then automates the design for network architecture using genetic algorithm operators, including (crossover, mutation, and selection), searching for a competitive architecture with few parameters. To validate the effectiveness of GA-UNet, we have conducted evaluations on three distinct image segmentation datasets: the lung segmentation dataset (lung segmentation) [19], the nuclei segmentation in a microscope images dataset (DSB 2018) [20], and the liver segmentation dataset (liver segmentation) [21]. Our experimental results show that the architectures proposed by the genetic algorithm, GA-UNet, succeed in reaching a U-Net architecture not only with a competitive performance against the manually designed U-Net [12], but they are even able to find a much shorter CNN architecture (0.24%, 0.48%, and 0.67% of the number of parameters in the original U-Net architecture for the lung image segmentation dataset, the DSB 2018 dataset, and the liver image segmentation dataset, respectively), which means reducing the computational effort without affecting their performance.

We also evaluated the performance of our proposed GA-UNet method for lung segmentation in comparison to several existing CNN-based techniques, such as [22,23,24,25]. The experimental results illustrates the outperformance of our method against these methods using multiple metrics such as the Dice similarity coefficient, precision, recall, and accuracy. The results underline the effectiveness of our proposed architecture over current state-of-the-art methods.

The rest of this work can be summarized as follows: Section 2 discusses the literature review. The proposed methodology is proposed in Section 3. Section 4 describes the materials of the experiments. The results are described in Section 5. Then, conclusions and limitations are drawn in Section 6. Finally, in Section 7, we discuss our future works.

## 2. Literature Review

### 2.1. U-Net Architecture

The advent of end-to-end fully convolutional networks (FCNs) involves a significant breakthrough by Ronneberger et al. [12]. Based on the concept of FCN, a novel framework called U-Net was proposed specifically for biomedical image segmentation. Today, U-Net has become one of the most frequently used FCNs in the medical image segmentation field. As mentioned in Section 1, the U-Net architecture is an encoder–decoder structure composed of two main paths. The first is the contracting path (encoder), consisting of repeated application of two successive 3 × 3 convolution layers with Rectified Linear Unit (ReLU) activation [26] and a max-pooling layer with two strides for down-sampling. In each step of down-sampling, the number of feature channels doubled. The second path is the expansive path (Decoder); every step consists of 2 × 2 up-convolution, which halves the number of feature channels, a concatenation between the cropped feature map from the corresponding layer in the contracting path and the up-sampling feature map, followed by two successive 3 × 3 convolution layers, each followed by Rectified Linear Unit (ReLU) activation. In the final stage of the network, an additional 1 × 1 convolution layer is used to reduce the feature map size from 64 to the required number of classes and produce the segmented image. The skip connections [17] are used to pass information from the encoder feature maps to the decoder part at the same level to compensate for the lost information. Figure 1a illustrates the original U-Net architecture.

The U-Net framework has played a significant role in generating advances in medical image segmentation, inspiring researchers to investigate novel directions and adapt the model to diverse imaging challenges. For instance, Ref. [27] introduced an innovative model called V-Net, which is specially crafted for the task of 3D medical image segmentation. Similarly, an innovative version of U-Net designed to work with 3D images was presented in a comprehensive study by [28]. In [29], a novel variant of U-Net called residual attention U-Net was presented for automatically segmenting COVID-19 chest CT images. This advanced deep learning model is based on the U-Net architecture, but it only employs both the residual network and attention mechanism to improve the feature extraction process, thereby producing superior multi-class segmentation results. The application of this method has led to a 10% enhancement in segmentation performance, which represents a significant advancement [25]. In further studies, Parcham et al. proposed a HybridBranchNet [30] to enhance the performance and efficiency of convolutional network models by optimally designing all dimensions, thereby increasing speed, reducing size and boosting accuracy optimization, while Fateh et al. in [31] introduced a model integrating language and digit recognition for multi-script images, leveraging transfer learning to improve image quality and recognition performance, with a particular emphasis on recognizing handwritten digits.

### 2.2. Genetic Algorithms

The genetic algorithm [32] is one of the famed metaheuristic optimization algorithms inspired by evolution through the process of natural selection. They are commonly used to solve optimization problems with high quality based on bio-inspired operators such as mutation, crossover, and selection. The process of GA consists of the following parts: initialization, selection, fitness evaluation, crossover, and mutation.

The genetic algorithm’s process starts with the random initialization of a population with a fixed number of individuals (each individual contains a set of variables called genes); the population will evolve by updating generations using the genetic operators: selection, crossover, and mutation. The selection operator is performed by selecting the fittest individuals and letting them survive for the next generation. During the crossover operator, a new population (known as the offspring) is produced by swapping the genes of each two individuals (known as parents). Next, in the mutation operator, an offspring is mutated by changing some of its genes with a low random probability in order to preserve the diversity and avoid premature convergence. Repeating the fitness evaluation, selection, crossover, and mutation operations will generate a new population with hopefully better solutions. When the stopping conditions are satisfied, the entire process terminates, and the last best individual throughout all generations is reported as the best solution. The basic steps of GA are summarized as follows:

1: Random initialization of population with a fixed number of individuals.

2: Evaluate the fitness of population.

3: Memorize the best solution.

4: **Repeat**

5:      Perform solution, crossover, and mutation.

6:      Fitness evaluation of the population.

7:      Update the best solution.

8:      Until the final criteria are met.

Genetic algorithms (GAs), inspired by the natural evolution process, have been applied to various medical image processing tasks. They are particularly effective in optimization tasks because of their robust search mechanisms. In [33], the authors adopted a unique optimization approach based on GA principles. The method was implemented for multilevel thresholding image segmentation, which is a task that involves the division of an image into various regions. Liu et al. [34] distinctively applied GA to optimize the hyperparameters of convolutional neural network (CNN) architecture for medical image denoising. Their objective was to refine the quality of medical images by reducing the noise and improving the clarity. The work of Nagarajan et al. in [35] demonstrated another implementation of GA in the medical imaging domain. They used GA to extract and select significant features from medical images. In [36], Mohamed et al. demonstrate the use of GA for edge-based segmentation applied to medical images. Finally, in [37], Sun et al. proposed an innovative methodology that leverages genetic algorithms (GA) for the automatic design of convolutional neural network (CNN) architectures. This approach, succinctly referred to as CNN-GA, demonstrates the ability to effectively identify the most optimal CNN structure to tackle image classification challenges.

As far as we know, no published papers have used genetic algorithms (GAs) to optimize convolutional neural network (CNN) architectures in the context of lung segmentation [19], cell nuclei segmentation (DSB 2018) [20] or liver segmentation [21] datasets.

## 3. Methodology

In this section, the details of the proposed GA-UNet method for lung segmentation [19], the DSB 2018 dataset [20], and liver segmentation [21] will be provided. We first describe the decoding strategy, define a specific search space and then realize the automated design for network architectures using genetic algorithm operations including crossover, mutation, and selection, searching for competitive architectures.

### 3.1. Search Space Encoding

To reduce the complexity of U-Net architecture, we adopted a block structure as shown in Figure 1b. We regrouped into one block all layers between two successive maxpooling or up-sampling layers. Therefore, each block Figure 2 consists of two 3 × 3 convolution layers with a Rectified Linear Unit (ReLU) activation function. The number of blocks is similar in the down-sampling path and the up-sampling path, with one single block in the bottleneck.

In this work, we will realize an automated design for network architecture by adjusting the number of blocks, their internal structure, and other parameters, using the genetic algorithm to find a satisfactory architecture. The first objective is to select the most relevant parameters that influence the performance of the discovered architecture. The following parameters are chosen to be optimized in this work:(1)Number of blocks ‘B’;(2)Number of filters ‘C’;(3)Filter size ‘F’;(4)Activation ‘A’;(5)Pooling ‘P’;(6)Batch-normalization ‘BN’;(7)Dropout ‘D’;(8)Optimizer ‘O’;(9)Learning rate ‘LR’;(10)Batch size ‘BA’.

We develop a simple yet effective encoding approach in which practically all the aforementioned parameters (B, C, F, A, P, BN, D, O, LR, and BA) are encoded as an integer vector. The block ‘B’ contains two operations: convolution and activation. The convolution operation has two parameters: the number of filters ‘C’ and the filter size ‘F’. The operations of activation, pooling, batch-normalization, and dropout are denoted by ‘A’, ‘P’, ‘BN’, and ‘D’, respectively. The optimizer, learning rate, and batch size are represented as ‘O’, ‘LR’, and ‘BA’.

When designing the search space, there are two main things to consider: making the search space concise but flexible, and limiting the number of architecture parameters.

Table 1 shows all the selected parameters, their codes, and range of values: (1) The number of blocks ‘B’ has a search space from (3 to 7), indicating the reproduced architecture can be with a minimum of three blocks to seven blocks as maximum. (2) The number of filters ‘C’ of the first block takes the range (2, 4, or 8 filters). (3) The value ranges of the filter size ‘F’ are from (3 to 7), where 3 represents the filter size (3 × 3), 4 represents (4 × 4), and so on, respectively, for each value. (4) The activation function ‘A’ takes the values ranges from (1, 2, or 3), representing ReLU [26], ELU [38], and LeakyReLU [39], respectively. (5) The type of pooling operation ‘P’ could take a value of (1) representing ’Maxpooling’ or (2) representing ‘Averagepooling’, all using the same stride of 2 × 2. (6) The batch-normalization operation ‘BN’ accepts a value (0 or 1) to indicate whether this operation should be used (see Figure 3b) or not (see Figure 3a). (7) Similar to the ‘BN’ operation, the dropout operation ‘D’ takes the value (0) or (1) to express if this operation could be used or not; in this work, the probability of the random drop is (0.3). (8) The search space of optimizer type ‘O’ is from (0 to 4), representing the SGD [40], RMSprop, Adam, and Adamax [41], respectively. (9) The learning rate ‘LR’ takes a value between $10^{-3}$ and $10^{-4}$. (10) The value range of batch size ‘BA’ is (4, 8, 16, or 32). During the genetic algorithm optimization, we use 20 epochs for the training phase of each candidate architecture.

The number of architecture-based parameters within this search space is significantly correlated by the number of blocks ‘B’, the number of filters ‘C’, and filter size ‘F’. Therefore, we have employed a range of values in our search space to confine the number of architecture-based parameters. Unlike the state-of-the-art medical image segmentation architectures, which may comprise millions of parameters, most of the network architectures within our search space have fewer than 0.6 million parameters.

With this coding scheme, the individuals of the genetic algorithm consist of 10 parameters (genes), whose structure is presented in Figure 4.

### 3.2. Evolutionary Algorithm

Our proposed approach GA-UNet is an iterative evolutionary technique designed to produce a progressively evolving population. The population comprises individuals, each of whom represents a distinct architecture. Each individual’s fitness is determined by evaluating the performance of the corresponding architecture in specific applications. Figure 5 and Algorithm 1 illustrate the proposed approach through a flowchart and pseudocode, respectively.

Algorithm 1 starts with a random initialization of a population $P_{0}$ of L individuals. After, we perform the number of generations G; at each generation, three evolutionary operations are applied (crossover, mutation, and selection) in order to improve the fitness of individuals. Next, the newly generated individuals are evaluated by training their corresponding architectures on the provided dataset. The process is carried out iteratively until the desired number of generations (G) is attained.
**Algorithm 1:** General framework of the proposed method
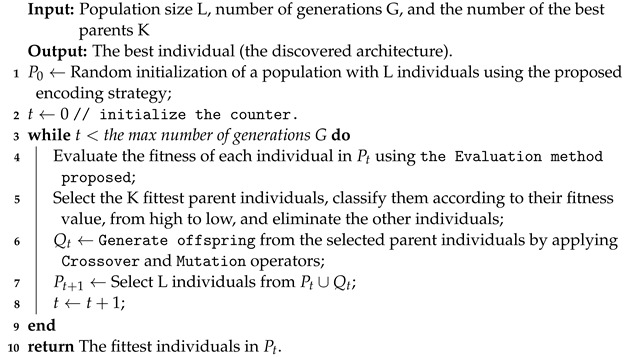


We begin by setting the population size L, the maximal number of generation G for the genetic algorithm, and K, the number of the best parent individuals. A population $P_{0}$ is initialized randomly with the predefined population size L, using the proposed encoding strategy (line 1). Then, we set a counter t for the current generation to zero (line 2); at each generation, we evaluate all the individuals of population $P_{t}$ by training their corresponding architectures on the given dataset (line 4). After that, we select the number of the best parents, K, based on their fitness and classify them according to their fitness values from high to low. Next, we applied a genetic operation such as crossover and mutation operators to generate a new offspring population $Q_{t}$ (line 6). Then, using the environmental selection method, we select from the current population $P_{t}$ a population $P_{t+1}$ of individuals surviving into the next generation (line 7). This population $P_{t+1}$ is composed explicitly of the K selected best parent individuals and the generated offspring population $Q_{t}$. Finally, the counter *t* is incremented by one (line 8), and the same process will be repeated until the counter exceeds the maximal number of generation G.

#### 3.2.1. Fitness Evaluation

To evaluate the fitness in this work, Algorithm 2 details the procedure for evaluating individuals in the population $P_{t}$. By giving a population $P_{t}$ containing L individuals, training data $D_{\mathrm{train}}$, and validation data $D_{\mathrm{valid}}$ in the beginning, F is set as an empty vector to memorize the fitness of all individuals (line 1). Algorithm 2 evaluates all the L individuals in the same manner (line 2→6) and transforms each individual of the population $P_{t}$ to its corresponding architecture. Then, we trained and validated the architecture on the training $D_{\mathrm{train}}$ and the validation data $D_{\mathrm{valid}}$, respectively. The performance of every architecture is evaluated, and its validation accuracy score $Acc_{\mathrm{valid}}$ will be calculated; the fitness value F is equal to $Acc_{\mathrm{valid}}$. At last, Algorithm 2 returns the population $P_{t}$ of L individuals with their fitness.
**Algorithm 2:** Fitness Evaluation
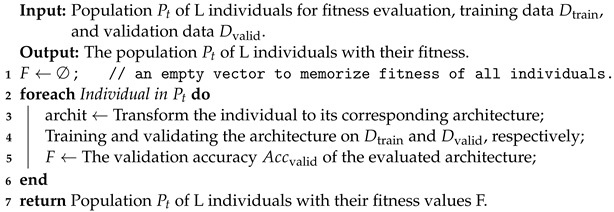


#### 3.2.2. Environmental Selection

Typically, genetic algorithms choose the next population by tournament or roulette selection. Both selection methods may miss the best individuals, which can lead to performance degradation [32]. This can have a significant impact on the optimization results. In contrast, if we select *K* parent individuals for the next generation, we have a higher probability of trapping the algorithm to a local optimum [18]. Algorithm 3 shows the details of the environmental selection. Firstly, given the current population $P_{t}$, the *K* best parent individuals of $P_{t}$ were selected and the generated offspring population, $Q_{t}$, become the next population $P_{t+1}$. Secondly, the $|P_{t}|-K$ individuals are selected from the generated offspring population $Q_{t}$ and then placed into $P_{t+1}$. The next population $P_{t+1}$ size is retained at the same as the current population $P_{t}$.
**Algorithm 3:** Environmental Selection
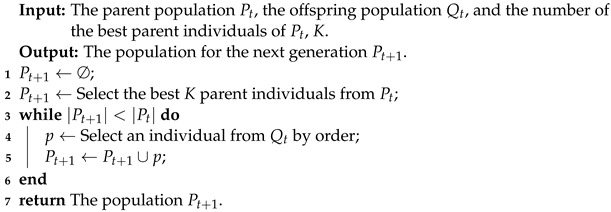


#### 3.2.3. Offspring Generating

In this work, we apply two genetic operators, crossover and mutation, to generate the offspring population $Q_{t}$. Algorithm 4 shows the process of generating offspring population $Q_{t}$, which contains two parts. The first is a crossover (lines 1 to 9), and the second is a mutation (lines 10 to 14). During the crossover part, the Algorithm 4 starts by selecting *K* best fitness parent individuals in population $P_{t}$ into $P_{\mathrm{best}}$. After that, two successive parent individuals from $P_{\mathrm{best}}$ are selected based on the better fitness score (line 4), and they exchange information with each other. This exchange can increase the performance of the algorithm. Otherwise, each parent individual is divided into two parts; these parts are swapped to create two new offspring (lines 5 to 8). In the second part, each offspring generated *p* experiences the mutation, and a random perturbation *r* is added to the last point *i* of the offspring *p*. This operation can increase the diversity of the population and prevent the algorithm from being trapped at the local optimum.
**Algorithm 4:** Offspring Generating
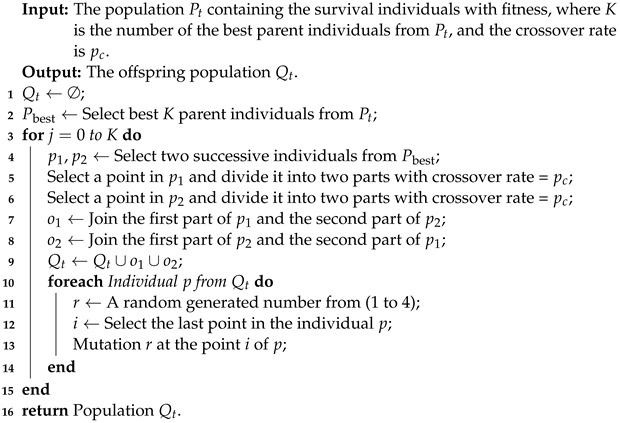


## 4. Materials for the Experiments

### 4.1. Datasets

Our work is evaluated on three datasets: The first is the lung segmentation dataset [19], which is a collection of CT images of manually segmented lungs and measurements in 2/3D provided by the Finding and Measuring Lungs in CT Data in Kaggle. These data consist of 267 2D images with a size of 512 × 512 pixels of each image. However, in this experiment, we just choose the 2D images and we resized the images to 256 × 256 pixels. For this study, 80% of the images were used for training and the remaining 20% were used for testing; 10% of the training data are used for validation. Each image comes with a corresponding fully annotated ground truth (GT) segmentation map for the lung (white) and other parts (black).

The second dataset contains nuclei segmentation in the microscopy images from the Data Science Bowl 2018 (DSB 2018) competition. DSB 2018 [20] is a collection of microscopy images of cell nuclei and their corresponding segmentation masks. The dataset consists of a training set and a test set. The training set contains 670 microscopy images, each with a corresponding segmentation mask. The test set contains 65 microscopy images without corresponding segmentation masks. The images were captured using fluorescence microscopy and are of varying sizes and shapes, with varying numbers and sizes of nuclei. For our experiments, we resized the images to 128 × 128 pixels. The segmentation masks are binary masks that indicate the location of each nucleus in the corresponding image. Among the 670 images, 80% of the samples were used for training, and the remaining 20% were used for testing; 10% of the training data are used for validation.

The third dataset, the liver dataset [21], is comparable to the first with the only difference being the replacement of the lung organ with liver. Additionally, the liver dataset has 400 additional images and 20 images validation of size 512 × 512 compared to the lung dataset. Moreover, we resized the images to 256 × 256 pixels, and the dataset will be divided as follows: 320 images used for the training set and 80 images for the test set. Both the first and the third datasets present similar challenges, as the images contain unclear edges and the organs in the images exhibit slight differences across different individuals. These challenges can impact the extraction of edges and the accurate localization of the organs that we aim to segment.

Before conducting our analysis, we carefully assessed the datasets used in this study and determined that the images contained within them exhibit minimal noise levels. Therefore, we have concluded that applying any preprocessing techniques to these images is unnecessary before proceeding with our experiments.

### 4.2. Loss Function

Image segmentation could be defined as a pixel-level classification task. An image is composed of multiple pixels grouped to represent different elements in the image. The choice of loss function is crucial when developing complicated image segmentation-based deep learning architecture, as they instigate the learning process of the algorithm [42]. In our case, using the lung segmentation dataset as example, the image is restricted to two classes at a pixel level: the lung class and non-lung class. Therefore, the problem is classifying these classes. For this purpose, the dice coefficient [43] is a function that measures the similarity of two sets and is one of the commonly used evaluation indicators in semantic segmentation. It has also been modified to be used as a loss function in this work as shown in Equation (Equation 1), which is given as:(1)$$\begin{array}{cc} \mathrm{Dice}\;\mathrm{loss}(X,Y) & =1-\frac{2|X\cap Y|}{\left|X|+|Y\right|} \end{array}$$

### 4.3. Evaluation Metrics

Lung segmentation is a binary classification problem that classifies each pixel in the lung CT image as either a lung or a non-lung tissue. Similar to cell nuclei segmentation from (DSB 2018) and liver segmentation, if a lung pixel is precisely classified, it is a True Positive (TP); if not, it is a False Positive (FP). If non-lung pixel is properly classified, it is a True Negative (TN); if not, it is a False Negative (FN).

To assess the performance of our segmentation models, we utilize multiple performance metrics for quantitative analysis. These metrics include accuracy (ACC), precision (PRE), recall (REC), dice coefficient (DSC), and Intersection over Union score (IoU). In addition, the area under the curve (AUC) is also used to evaluate the performance of the proposed approaches against the original U-net network. The calculation of the overall accuracy is performed using Equation (Equation 2), while the calculation of precision is performed using Equation (Equation 3).
(2)$$\begin{array}{cc} \mathrm{ACC} & =\frac{TP+TN}{TP+TN+FP+FN} \end{array}$$
(3)$$\begin{array}{cc} \mathrm{PRE} & =\frac{TP}{TP+FP} \end{array}$$

In addition, recall is computed using the following Equation (Equation 4):(4)$$\begin{array}{cc} \mathrm{REC} & =\frac{TP}{TP+FN} \end{array}$$

According to [43], the dice coefficient can be expressed as shown in Equation (Equation 5), where *X* represents the ground truth, and *Y* represents the segmentation result.
(5)$$\begin{array}{cc} \mathrm{DSC} & =\frac{2|X\cap Y|}{\left|X|+|Y\right|} \end{array}$$

The formula for the Intersection over the Union score is given by Equation (Equation 6) as described in [44].
(6)$$\begin{array}{cc} \mathrm{IoU} & =\frac{\left|X\cap Y\right|}{\left|X\cup Y\right|} \end{array}$$

## 5. Experiments and Results

In this study, the experiments comprise two distinct phases. The first phase focuses on using genetic algorithms to explore neural architectures, while the second phase centers on training and validating both the original U-net architecture [12] and newly discovered architectures from scratch to demonstrate their performances on the three datasets: the lung segmentation dataset [19], DSB 2018 dataset [20], and liver segmentation dataset [21]. In this section, we will present a detailed overview of the two experimental phases as well as a comprehensive analysis of the results obtained from this study.

### 5.1. Experimental Setup

The proposed architectures and all other experiments were implemented in Python 3.9, using the Keras and TensorFlow frameworks as the backend. All Python scripts were executed on an HPC cluster server, named “HPC-MARWAN”, which was equipped with an Nvidia Tesla P100 GPU with 12 GB of memory and 192 GB total system memory.

Genetic algorithm parameters:

In the architecture search phase, we use the range of values shown in Table 1 to design U-Net architectures automatically, and the satisfactory architecture can be discovered by optimizing this range of values using the genetic algorithm over a certain number of generations. However, the main parameters used to run the genetic algorithm in the first phase are shown in Table 2.

During the second stage, the training configurations were kept consistent with the architecture search stage, such as the loss function and data splitting for all three datasets (lung segmentation, cell nuclei segmentation (DSB 2018), and liver segmentation). Nevertheless, the most notable change was an increase in the number of training epochs, which was necessary to ensure model convergence.

### 5.2. Experimental Results

Our study is centered around the U-shaped encoder–decoder structures, which provide the foundation for our work. To evaluate the effectiveness of the discovered architectures generated by our method GA-UNet, we conducted a comparative analysis against the original U-Net [12] model, which is a widely used U-shaped model that serves as the baseline in our study. To ensure a fair and accurate comparison between the original U-Net [12] and the discovered architectures, we trained the former using the same parameter settings as each discovered architecture.

#### 5.2.1. Discovered Architecture for Lung Segmentation Dataset

The best architecture generated from the lung segmentation dataset [19] after applying our approach had the following structure: five convolutional blocks, each block containing two convolution layers (see Figure 6). The first block had eight convolutional filters and 5 × 5 size of filters, with “ELU” activation, no batch normalization, and no dropout regularization. The max-pooling layer had a size of 2 × 2. The optimizer used was Adam with a learning rate of $10^{-4}$ and a batch size of 8.

In this experiment, the discovered model was trained for 100 epochs, and its performance was subsequently evaluated against that of the baseline model, U-Net [12]. Figure 7 displays the accuracy and loss plots of the discovered architecture and the original U-Net [12] for the two training and validation phases. The first two plots (a) and (b) in Figure 7 show the accuracy of training and validation data during the training of the models for the lung segmentation dataset; the two graphs show that the discovered model performs well during the training and validation phases compared to the original U-Net [12]. The accuracy graphs of the GA-UNet and U-Net [12] models showed a significant increase during the first 20 epochs and 15 epochs, respectively, after which the rate of improvement slowed down and eventually became nearly constant. Furthermore, the loss for both training and validation data during the training of the models for the lung segmentation dataset are shown in the graphs (c) and (d) in Figure 7. These graphs show that the discovered model has good performance. It can be seen that the loss of the discovered model for both phases, training and validation, significantly decreased during the initial 30 epochs, which is followed by a period of relative stability. Based on the information presented in the four graphs of Figure 7, it can be concluded that the model designed by our proposed method performs well in accurately segmenting the lung from abdominal CT scan images.

The quantitative analysis of the proposed method for the lung segmentation dataset and the comparison against the original U-Net [12] model are presented in Table 3. It shows that the discovered architecture produced by our GA-UNet method outperforms the original U-Net [12] model in terms of computational efficiency, and it delivers a highly competitive performance compared to the original U-Net [12] model. The number of parameters in GA-UNet is approximately 0.24% of the number of parameters in the original U-Net [12]. This significant reduction in parameters results in a much smaller model size, which is approximately 0.28% of the size of the original U-Net [12]. Additionally, GA-UNet has a much shorter training time, which is approximately 15% of the training time required for the original U-Net [12]. Both GA-UNet and the original U-Net [12] model achieved high-performance scores across all metrics. In terms of DSC and IoU, the U-Net method has a slightly higher score than the GA-UNet method (0.9843 vs. 0.9812 and 0.9691 vs. 0.9632, respectively). However, the GA-UNet method achieved a higher AUC score than U-Net [12] (0.9922 vs. 0.9889), indicating its ability to better distinguish between positive and negative cases. The PRE and REC scores for both methods are also high, indicating their ability to detect most of the relevant features in the images. The REC score of GA-UNet is slightly higher than that of U-Net [12] (0.9915 vs. 0.9932), indicating its ability to identify more of the lung region in CT scan images. However, the PRE score of U-Net [12] is slightly higher than that of GA-UNet (0.9777 vs. 0.9726), indicating its ability to produce fewer false positives.

Overall, the results suggest that both methods are highly effective for lung segmentation tasks. However, the GA-UNet method shows a slightly better performance in terms of AUC score, while the U-Net [12] method shows a slightly better performance in terms of DSC and IoU scores.

For the qualitative analysis, we show some precise and promising segmentation results in Figure 8 to visually compare our method with the original U-Net [12] model. The lung segmentation result depicted in Figure 8 is deemed satisfactory, as the proposed approach GA-UNet was able to almost completely segment the lung from the CT scan images. But in some instances, as shown in rows 2 and 4, it missed a small portion of the lung.

##### Comparison with Existing Methods

For an extensive study, we compare the test results from our discovered architecture for the lung segmentation dataset to those of the other existing methods (specifically CNN-based techniques). The experimental results have been collected and presented concisely in Table 4. In this table, the results of the existing methods have been derived from [25]. In comparison to the current state-of-the-art methods, our proposed GA-UNet method significantly outperforms the others. The Dice similarity coefficient (DSC) of our GA-UNet stands at 0.9812, and it clearly outperforms ResBCDU-Net [25] (0.9715), BCDU-Net [24] (0.9632), ResNet34-Unet [23] (0.9528) and RU-Net [22] (0.9493). It is worth noting that although the precision (PRE) of GA-UNet is slightly lower than some methods (0.9726 compared to the highest, ResBCDU-Net [25] 0.9912), it significantly surpasses all in terms of recall (REC) with a score of 0.9915 and accuracy (ACC) with a score of 0.9878, this is compared to the second highest in these categories, BCDU-Net [24], which scored 0.9803 in recall, and ResBCDU-Net [25], which scored 0.9758 in accuracy.

#### 5.2.2. Discovered Architecture for DSB 2018 Dataset

The best architecture generated from the cell nuclei segmentation in the DSB 2018 dataset [20] after applying our approach had the following structure: five convolutional blocks, each block containing two convolution layers (see Figure 9). The first block had eight convolutional filters and 7 × 7 size of filters, with “ReLU” activation and batch normalization operation, no dropout regularization. The max-pooling layer had a size of 2 × 2. The optimizer used was Adamax with a learning rate of $5\times 10^{-4}$ and a batch size of 16.

In this experiment, the model discovered by our approach GA-UNet was trained for 100 epochs, which was followed by a performance evaluation against the baseline model, U-Net [12]. Figure 10 depicts the accuracy and loss plots for the discovered architecture and the original U-Net [12] during the training and validation phases. Specifically, plots (a) and (b) in Figure 10 show the training accuracy and the validation accuracy for the DSB 2018 dataset. These graphs highlight that the discovered architecture exhibited competitive performance during training and validation phases compared to the original U-Net [12]. In addition, Figure 10 displays the loss graphs (c) and (d) for both the training and validation data during the model training for the DSB 2018 dataset. These graphs indicate that the loss plots for the original U-Net [12] and the loss plots for GA-UNet exhibit a consistent and gradual decrease in both the training and validation loss throughout the 100 epochs of training. This results indicates that the models continued to learn and improve, although they needed more training epochs to attain convergence.

Table 5 presents a quantitative analysis of the proposed approach for the DSB 2018 dataset, including a comparison with the original U-Net [12] model. It is clear that the GA-UNet model is significantly more computationally efficient than the original U-Net model [12]. The proposed approach GA-UNet has approximately 99.52% fewer parameters than the original U-Net [12], resulting in a model size that is only 0.54% of the size of the original U-Net [12]. Moreover, GA-UNet requires only about 19% of the training time required for the original U-Net [12]. Both U-Net [12] and GA-UNet models performed well on the DSB 2018 dataset, with U-Net [12] showing slightly higher performance on most metrics, including DSC, PRE, REC, IoU, ACC. The U-Net [12] model had a DSC of 0.9197, PRE of 0.9262, REC of 0.9141, IoU of 0.8514, and ACC of 0.9635. In contrast, the GA-UNet model achieved a DSC of 0.9018, PRE of 0.9199, REC of 0.9041, IoU of 0.8213, and ACC of 0.9596. However, GA-UNet had a significantly higher area under the curve (AUC) score of 0.9836 compared to U-Net’s AUC score of 0.9518; this suggests that GA-UNet is better at distinguishing between positive and negative pixels.

The qualitative analysis was conducted by comparing nuclei segmentation of the DSB 2018 dataset [20], regarding outcomes obtained from the proposed GA-UNet approach with those from the original U-Net [12] model, as presented in Figure 11. This figure provides a clear visual representation of the results. It is apparent from Figure 11 that the GA-UNet approach achieved near-complete segmentation of the cell nuclei of microscopy images, indicating satisfactory results. Nevertheless, in some cases (rows 2 and 6), small portions of the nuclei were missed, highlighting areas for potential improvement.

#### 5.2.3. Discovered Architecture for Liver Segmentation Dataset

The best architecture generated from the liver segmentation dataset [21] after applying our approach had the following structure: seven convolutional blocks, each block containing two convolution layers (see Figure 12). The first block had eight convolutional filters and filters of 4 × 4 size “ReLU” activation and batch normalization operation with no dropout regularization. The max-pooling layer had a size of 2 × 2. The optimizer used was Adam with a learning rate of $10^{-4}$ and a batch size of 4.

In this experiment, the discovered model was trained for 100 epochs, and its performance was subsequently evaluated against that of the baseline model, U-Net [12].

Table 6 provides the quantitative analysis results of our proposed approach for the liver segmentation dataset as well as a comparison with the original U-Net [12] model. From these results, it is evident that the GA-UNet model continues to demonstrate impressive computational efficiency and competitive segmentation performance against the original U-Net model [12]. In terms of performance metrics, both models have high values for the Dice similarity coefficient (DSC), precision (PRE), recall (REC), and area under the receiver operating characteristic curve (AUC). However, the proposed GA-UNet model has slightly lower values for all of these metrics compared to the U-Net model except for the AUC, where it performs slightly better. For DSC, PRE, and REC, the original U-Net [12] model has a slightly higher score (0.9886, 0.9892, and 0.9952) than the GA-UNet method (0.9788, 0.9837, and 0.9923). Regarding IoU, the original U-Net [12] model has a higher score (0.9775) compared to the GA-UNet model (0.9584). This indicates that the original U-Net [12] has a higher intersection between the predicted segmentation and the ground truth. Finally, both models have the same accuracy ACC (0.9858), but the GA-UNet model has a higher AUC score (0.9869) than the original U-Net [12] model (0.9736). This means that the proposed GA-UNet can better distinguish between positive and negative instances in the image for the liver segmentation dataset.

It is important to note that the proposed GA-UNet model achieves these high-performance metrics while also having a much shorter training time compared to the original U-Net model [12]. The proposed approach, GA-UNet, has approximately 99.33% fewer parameters than the original U-Net [12], resulting in a model size of only 0.75% of the size of the original U-Net [12]. Additionally, GA-UNet has a much shorter training time, taking only about 29% of the time required for the original U-Net [12].

In the qualitative analysis, Figure 13 displays the liver segmentation outcomes obtained from our proposed approach, GA-UNet, compared to those from the original U-Net [12] model, providing a clear visual representation of the results. As demonstrated in Figure 13, the liver segmentation results are considered satisfactory, given that GA-UNet achieved near-complete segmentation of the livers in the scan images. However, as observed in rows 2 and 3, there were instances where a small portion of the liver was missed. On the other hand, in row 6, GA-UNet exhibited better liver segmentation performance than the original U-Net.

## 6. Conclusions

This paper proposes an automated architecture design approach based on the U-shaped encoder–decoder structure called GA-UNet. This approach utilizes the genetic algorithm (GA) to generate competitive architectures with a restricted number of parameters through a compact but dynamic search space. We evaluated the performance of GA-UNet on three datasets: lung segmentation, cell nuclei segmentation (DSB 2018), and liver segmentation datasets.

The experimental findings indicate that the discovered architectures generated by our proposed approach, GA-UNet, achieve competitive performance compared with the baseline model, such as U-Net [12], with a considerable reduction in parameter numbers and computation complexity. It requires only 0.24%, 0.48% and 0.67% of the number of parameters present in the original U-Net architecture for the lung image segmentation dataset, the DSB 2018 dataset and the liver image segmentation dataset, respectively; this translates into a reduction in computational effort without compromising on performance, which makes our proposed methodology, GA-UNet, more suitable for deployment in resource-limited environments or real-world implementations. We further examined the performance of our proposed GA-UNet method for lung segmentation by contrasting it with several existing CNN-based techniques, such as those detailed in [22,23,24,25]. The empirical findings demonstrate, via multiple metrics, the outperformance of our approach in comparison to these techniques.

Despite these promising results, there are a few limitations to note. Firstly, the GA-UNet performance may depend on the initial parameters of the genetic algorithm, such as population size and number of generations, which we intend to adjust in future research. Secondly, the proposed search space design might not capture all potential architectural solutions, needing further investigation.

## 7. Future Works

For future works, we intend to expand the size of the population and the number of generations in our GA to investigate whether further improvements can be achieved in performance; we will also focus on exploring a more practical search space design. In addition, to further confirm the effectiveness and generality of the proposed approach, we should expand its application to new datasets for different applications in the field of medical imaging.

## Figures and Tables

**Figure 1 jpm-13-01298-f001:**
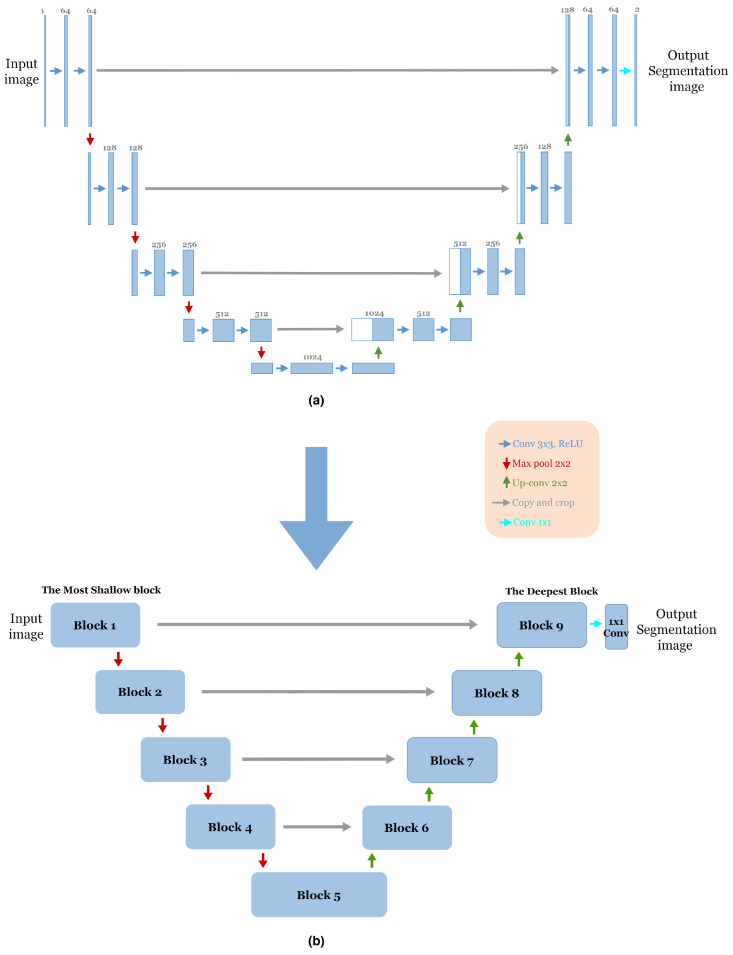
(**a**) The Original U-Net Architecture [12]. (**b**) The adopted block structure of the original U-net architecture [12].

**Figure 2 jpm-13-01298-f002:**
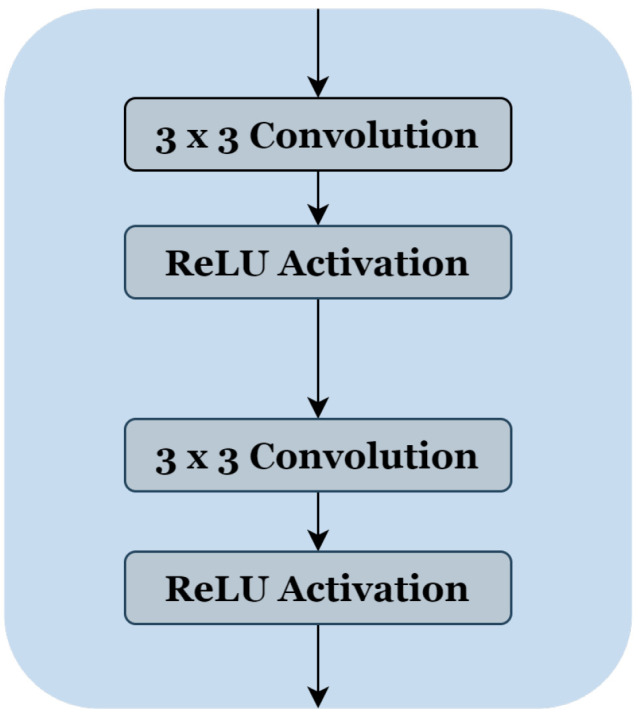
The block’s internal structure in the original U-Net architecture [12].

**Figure 3 jpm-13-01298-f003:**
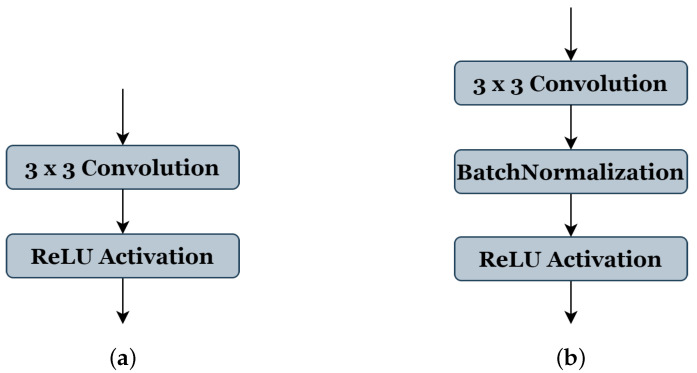
Convolution and activation layers in the original U-Net model (**a**), and batch-normalization layer implementations in our method (**b**).

**Figure 4 jpm-13-01298-f004:**
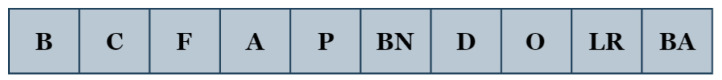
Encoding structure of the proposed GA-UNet.

**Figure 5 jpm-13-01298-f005:**
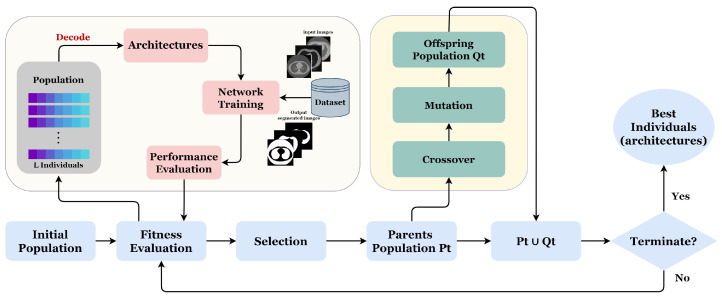
Overall framework of the proposed method.

**Figure 6 jpm-13-01298-f006:**
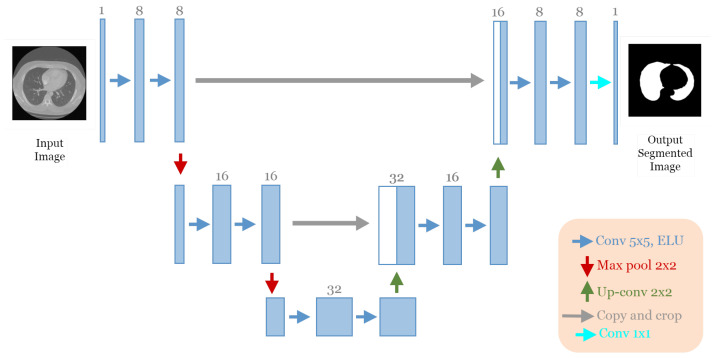
The discovered architecture generated by the proposed GA-UNet for the lung segmentation dataset.

**Figure 7 jpm-13-01298-f007:**
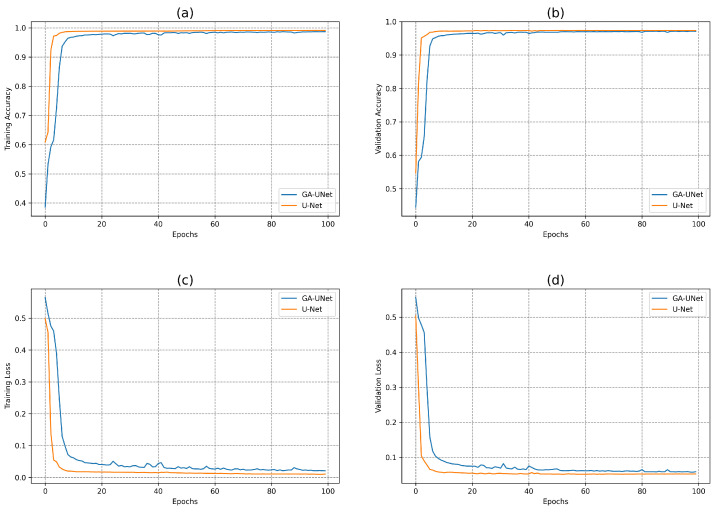
Accuracy and loss plots during training and validation phases of the proposed GA-UNet and the original U-Net [12] for the lung segmentation dataset. (**a**) Training accuracy plots, (**b**) Validation accuracy plots, (**c**) Training loss plots, and (**d**) Validation loss plots.

**Figure 8 jpm-13-01298-f008:**
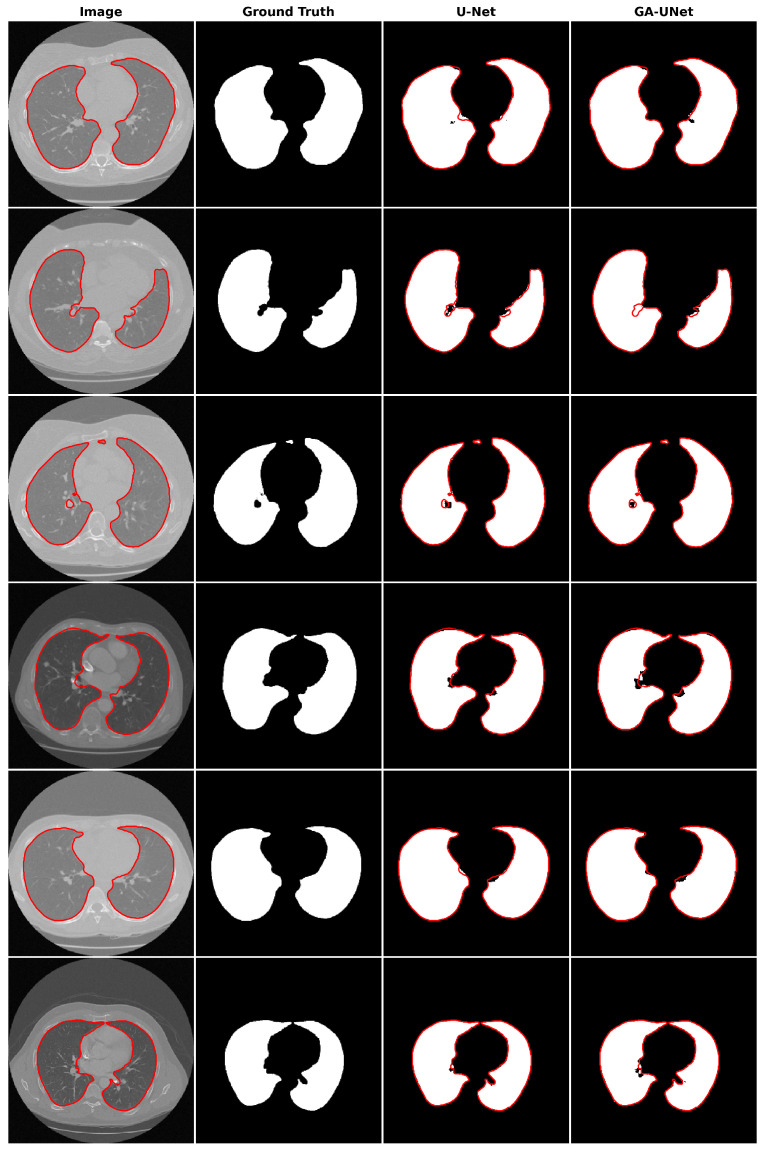
Segmentation results on lung segmentation dataset. From left to right, the columns represent in order: the input image, ground truth, U-Net prediction, and GA-UNet prediction. The red curves represents the actual area of the lung.

**Figure 9 jpm-13-01298-f009:**
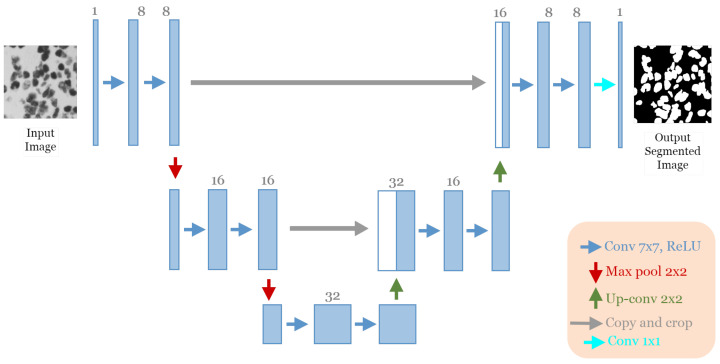
The discovered architecture generated by the proposed GA-UNet for the DSB 2018 dataset.

**Figure 10 jpm-13-01298-f010:**
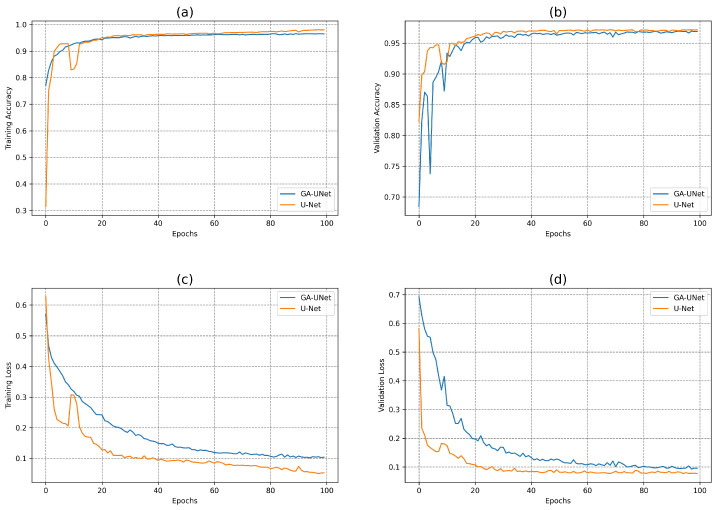
Accuracy and loss plots for the training and validation phases of the proposed GA-UNet and the original U-Net [12] for the DSB 2018 dataset. (**a**) Training accuracy plots, (**b**) Validation accuracy plots, (**c**) Training loss plots, and (**d**) Validation loss plots.

**Figure 11 jpm-13-01298-f011:**
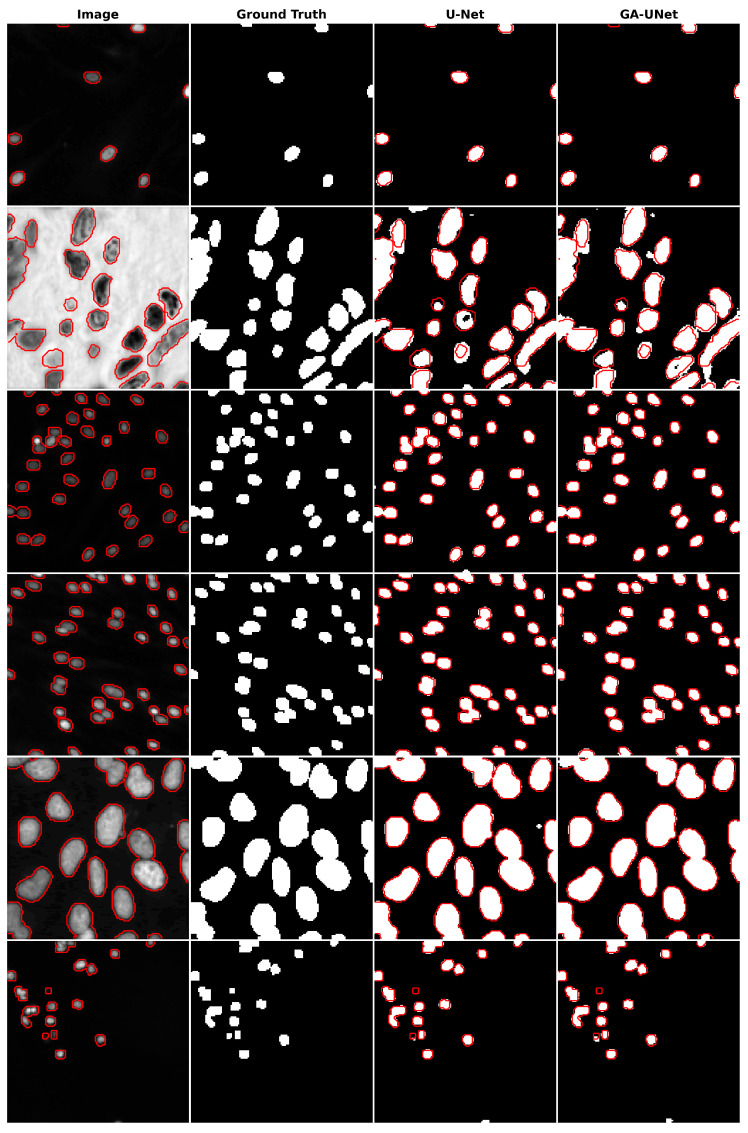
Segmentation results on nuclei segmentation of DSB 2018 dataset. From left to right, the columns represent in order: the input image, ground truth, U-Net prediction, and GA-UNet prediction. The red curves represents the actual area of the cell nuclei.

**Figure 12 jpm-13-01298-f012:**
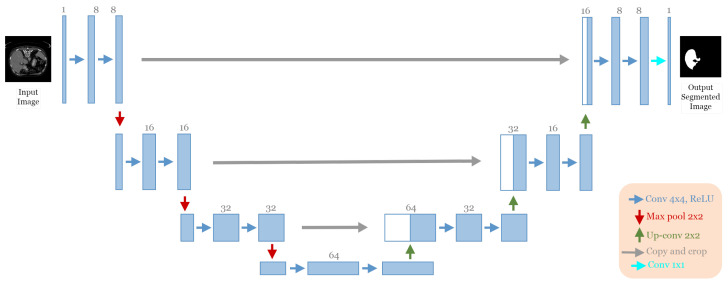
The discovered architecture generated by the proposed GA-UNet for the liver segmentation dataset.

**Figure 13 jpm-13-01298-f013:**
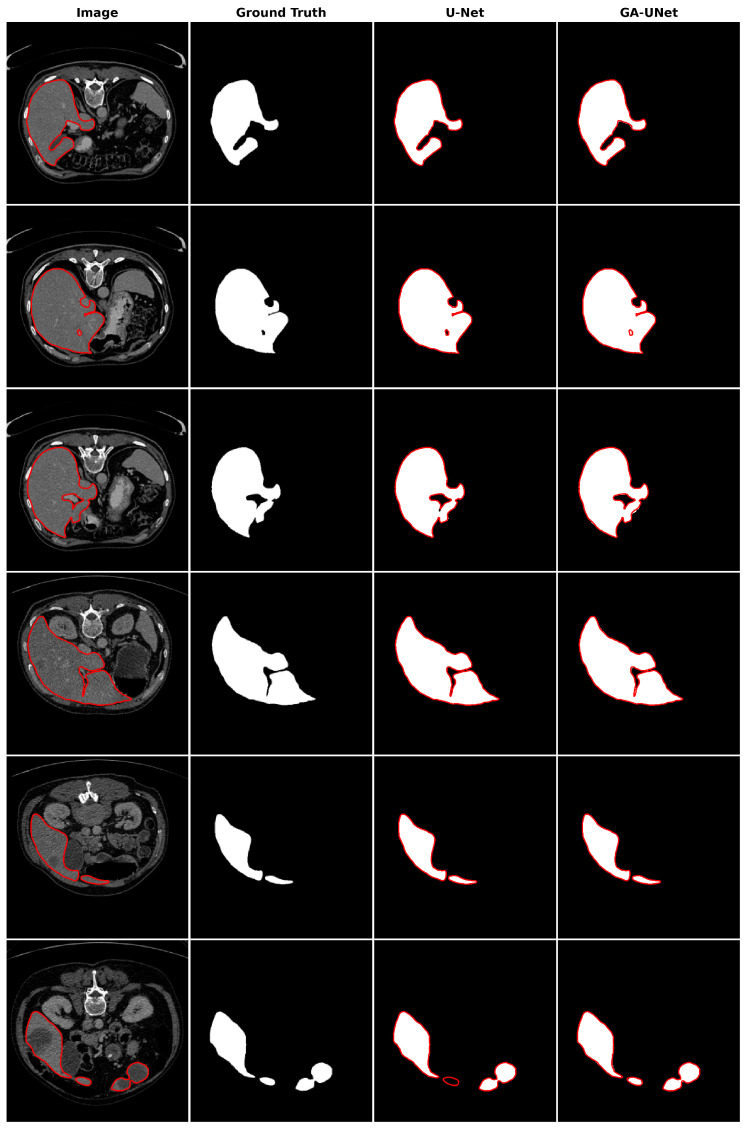
Segmentation results on liver segmentation dataset. From left to right, the columns represent in order: the input image, ground truth, U-Net prediction, and GA-UNet prediction. The red curves represents the actual area of the liver.

**Table 1 jpm-13-01298-t001:** List of the selected parameters to be optimized, their codes, and range of values.

Numbers	Parameters	Code	Range of Values
1	Number of blocks	‘B’	[3, 7]
2	Number of filters	‘C’	2, 4, 8
3	Filter size	‘F’	[3, 7]
4	Activation	‘A’	1, 2, 3
5	Pooling	‘P’	1, 2
6	Batch normalization	‘BN’	0, 1
7	Dropout	‘D’	0, 1
8	Optimizer	‘O’	1, 2, 3, 4
9	Learning rate	‘LR’	[$10^{-3},10^{-4}$]
10	Batch size	‘BA’	4, 8, 16, 32

**Table 2 jpm-13-01298-t002:** Genetic algorithm parameters.

Parameters	Value
Number of generations	5
Population size	10
Number of best parent individuals K	5
Crossover rate	0.5
Mutation rate	0.1

**Table 3 jpm-13-01298-t003:** Experimental results of the proposed approach for the lung segmentation dataset and comparison against the original U-Net model.

Methods	Year	Params	Model Size	DSC	PRE	REC	IoU	ACC	AUC	Training Time
U-Net [12]	2015	31.03 M	363.8 MB	0.9843	0.9777	0.9932	0.9691	0.9890	0.9889	00:15:20
Proposed GA-UNet	2023	0.076 M	1.02 MB	0.9812	0.9726	0.9915	0.9632	0.9878	0.9922	00:02:23

**Table 4 jpm-13-01298-t004:** Comparison of the proposed approach against the existence methods on the lung segmentation dataset.

Methods	Year	DSC	PRE	REC	ACC
RU-Net [22]	2018	0.9493	0.9552	0.9721	0.9715
BCDU-Net [24]	2019	0.9632	0.9902	0.9803	0.9721
ResNet34-Unet [23]	2020	0.9528	0.9732	0.9835	0.9673
ResBCDU-Net [25]	2021	0.9715	0.9912	0.9701	0.9758
Proposed GA-UNet	2023	0.9812	0.9726	0.9915	0.9878

**Table 5 jpm-13-01298-t005:** Experimental results of the proposed approach for the DSB 2018 dataset and comparison against the original U-Net model.

Methods	Year	Params	Model Size	DSC	PRE	REC	IoU	ACC	AUC	Training Time
U-Net [12]	2015	31.03 M	363.8 MB	0.9197	0.9262	0.9141	0.8514	0.9635	0.9518	00:12:24
Proposed GA-UNet	2023	0.148 M	1.95 MB	0.9018	0.9199	0.9041	0.8213	0.9596	0.9836	00:02:23

**Table 6 jpm-13-01298-t006:** Experimental results of the proposed approach for liver segmentation dataset and comparison against the original U-Net model.

Methods	Year	Params	Model Size	DSC	PRE	REC	IoU	ACC	AUC	Training Time
U-Net [12]	2015	31.03 M	363.8 MB	0.9886	0.9892	0.9952	0.9775	0.9858	0.9736	00:22:24
Proposed GA-UNet	2023	0.207 M	2.73 MB	0.9788	0.9837	0.9923	0.9584	0.9858	0.9869	00:06:24

## Data Availability

Not applicable.

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
