# Peer review of "Medical Image Segmentation Using Automatic Optimized U-Net Architecture Based on Genetic Algorithm"

_jpm, 2023, doi:10.3390/jpm13091298_

Round 1
Reviewer 1 Report
The GA-UNet method offers an innovative automated design approach for convolutional neural network (CNN) architecture with a U-shaped topology, specifically tailored for medical image segmentation. By leveraging genetic algorithms, the model efficiently generates the architecture and intelligently selects its hyperparameters. Rigorously evaluated on three distinct image segmentation datasets, GA-UNet demonstrates competitive performance while maintaining a significantly reduced architecture size compared to manually designed U-Net models. This enhanced efficiency renders GA-UNet highly suitable for resource-limited environments and real-world applications, where rapid and efficient inference is imperative.
Weak points and suggestions:
1) The abstract of appears weak and warrants a complete rewrite. It lacks essential information, including the best accuracy achieved by the proposed model. To strengthen the abstract, the authors should clearly state their research objectives, methodology, key findings, and the highest accuracy attained by their model. A well-crafted abstract will provide readers with a concise yet comprehensive overview of the paper, enticing them to delve deeper into the research.
2) In the introduction, it is essential for the authors to highlight their novelties in a separate paragraph, boldly presenting the unique contributions of their research. This distinct emphasis will allow readers to quickly grasp the groundbreaking aspects of the GA-UNet method.
3) The “Literature review” section would benefit from expansion, including a more comprehensive discussion of relevant methods. By incorporating additional approaches and their respective findings, the authors can provide readers with a deeper understanding of the existing research landscape. This would help highlight the distinctive contributions of the GA-UNet method in the context of medical image segmentation, showcasing its potential advantages and impact in comparison to established techniques.
4) The “Literature review” section should be extended to include new methods in the state of the art. There are some deep-learning approaches that is suggested to add:
-- “HybridBranchNet: A novel structure for branch hybrid convolutional neural networks architecture” Neural Networks 165 2023
-- “Multilingual handwritten numeral recognition using a robust deep network joint with transfer learning” Information Sciences 581 (2021)
5) The presentation of Figure 5, illustrating the overall framework of the Proposed Method, appears to be lacking in quality and clarity. The authors should focus on improving the figure to enhance its visual appeal and effectively convey the essential components of their approach. A clearer representation of the framework will significantly contribute to the readers' understanding and appreciation of the proposed method.
6) In the "Experiments and results" section, the comparison of the proposed method solely with the original U-Net is insufficient. To provide a more comprehensive evaluation, the authors should include a comparison with other recent U-Net-based approaches. For instance, one such method could be “ResBCDU-Net: A Deep Learning Framework for Lung CT Image Segmentation” (Sensors 2021). This additional comparison will strengthen the paper's findings and establish the effectiveness of the proposed method in relation to other state-of-the-art U-Net-based techniques.
7) The conclusion could be strengthened by summarizing the key findings and contributions of the research in a more concise and clear manner. Additionally, the implications and limitations of the research could be discussed in more detail.
8) The language of this paper requires further polishing.
-
Reviewer 2 Report
The work is good and organized and needs some evaluation:
1. In the abstract, the authors should refer to the most important results of the systems.
2. The main contributions should be shown clearly at the end of the introduction section.
3. A Figure should be made for the entire study methodology, from entering the images to the result of the systems.
4. The performance of the systems should be discussed and compared with relevant recent studies.
5. What are the future work of the study, and what are the limitations faced by the authors?
6. the importance of segmentation of medical images and their purpose should be written.
Minor editing of English language required
Round 2
Reviewer 1 Report
The authors have satisfactorily addressed all my concerns in the revised manuscript, and I recommend accepting it for publication.
Reviewer 2 Report
Accept in present form
Minor editing of English language required